# Spatial Correlation Network Structure of Carbon Emission Efficiency of Railway Transportation in China and Its Influencing Factors

**Ningxin Zhang** **, Yu Zhang and Hanli Chen \***

School of Civil Engineering, Central South University, Changsha 410083, China; 214812379@csu.edu.cn (N.Z.); 204812389@csu.edu.cn (Y.Z.)
\* Correspondence: chenhanli@csu.edu.cn; Tel.: +86-137-8721-0114

**Abstract:** Railway carbon emissions reduction is of great significance. In this study, carbon emission efficiency in railway transportation in China's 31 provinces is measured for 2006–2019 based on an unexpected output slack-based measure (SBM) model. A gravity matrix of the spatial correlation network for carbon emission efficiency is constructed using the modified gravity model, the spatial network structure is explored using social network analysis, and the factors influencing the spatial network are analyzed using the quadratic assignment procedure (QAP) model. Based on the results, several conclusions can be drawn: (1) the carbon emissions efficiency of railway transportation in China increased periodically during the study period, but there are still great differences between regions. (2) The carbon emission efficiency in railway transportation shows significant characteristics of spatial correlation networks. (3) The inter-provincial associations gradually increased, while there are still large regional differences in the spatial correlation network. (4) Differences in spatial adjacency, economic development and scientific and technological advancement have significant positive impacts on the spatial correlation network. This research will help policy makers formulate relevant policies to promote the regional coordinated development of low-carbon railway transportation.

**Keywords:** railway carbon emission efficiency; spatial network; SBM; spatial network structure; social network analysis; QAP model

## 1. Introduction

At the 75th United Nations General Assembly, China committed to achieving a carbon peak by 2030 and carbon neutrality by 2060 [1]. The constant development of the national economy in China has accelerated the circulation of labor, technology, capital, products, and other elements of the economic market, with transportation becoming an important supporting factor for social and economic development [2,3]. The development of the transportation industry has resulted in a rapid increase in the volume of carbon emissions. China's transportation industry accounted for 14.82% of the world's total transportation carbon emissions in 2019, second only to 20.81% of the United States [4]. Hence, the carbon emission reduction of the transportation industry is crucial to the realization of the carbon peak and carbon neutrality.

Rail is a significant mode of transport. The scale of railway transportation in China is increasing rapidly. China's railway mileage in operation exceeded 150,000 km in 2021, and will reach 165,000 km in 2025 according to the 14th Five-Year Plan for railway development. This extension of the railway scale produces more carbon emissions. Additionally, research has been conducted to show that carbon emissions from other industries (industry, agriculture, fishery, animal husbandry, forestry, construction, trade and services, and transportation other than railway) are highly sensitive to those from the railway transport. That is, while railway transport reduces carbon emissions by one unit, other sectors tend to reduce total carbon emissions by more than one unit [5]. This indicates that the emission

reduction of the railway transportation industry is of great significance to the whole national economic sector. As can be seen from the above, railway carbon emissions reduction is of great significance to the low-carbon development of the transport industry and the achievement of China's carbon emissions reduction target.

However, the carbon emission reduction capacity of railway transportation measured only by carbon emissions is unilateral. To determine the carbon emission reduction capacity of railway transportation, it is important to consider the economic output it can generate while producing carbon emissions, as well as the amount of carbon emissions that the economic output will generate [6]. Therefore, improving the carbon emission efficiency of railway transportation is key to improving its ability to reduce carbon emissions.

Measuring carbon emissions from railway transportation forms the basis for studying its carbon emission efficiency. Evidence suggests that its carbon dioxide emissions can be calculated based on the number of different types of energy consumed by it and the carbon emission factors of these types of energy [5], which is a "top-down" approach. However, such methods may make it difficult to unify the carbon emission coefficients due to regional problems. Therefore, some scholars propose the standardization of carbon emission coefficients of different fuels [7]. To do this, they convert different types of energy consumption into the consumption of standard coal and calculate carbon emissions according to the carbon emission coefficient of standard coal [8,9]. In addition to measuring carbon emissions based on energy consumption, it is also possible to calculate energy consumption using various metrics associated with the operating volume of different transport vehicles, such as operating mileage [10], transport turnover [11,12], and the number of engines [13]. Determining energy consumption per unit of operating volume and using this value to estimate carbon emissions is called the "bottom-up" method. Compared with the "top-down" method, the "bottom-up" method does not require total energy consumption data and can be calculated by using the operation data of transport vehicles.

Extensive research has been conducted on the conceptual definition, efficiency measurement, influencing factors, and spatial analysis of the carbon emission efficiency of railways. Carbon emission efficiency is an index reflecting the input–output relationship of production activities on the premise of carbon emission. In general, the ratio of the minimum carbon emission that can be achieved in theory to the actual carbon emission under a certain input and output is known as carbon emission efficiency [4]. With limited input factors, higher carbon emission efficiency in railway transportation leads to higher economic output or lower carbon emissions, which promote regional economy and low carbon development. It is evident that carbon emission is an undesired output.

When measuring carbon emission efficiency in railway transportation, carbon emissions must be included in the input–output model. This can be achieved in two ways: by converting carbon emissions into input variables, or by including carbon emissions as unintended outputs in the research system. The latter approach aligns with the concept of carbon efficiency more closely. At present, the carbon emission efficiency measured using carbon emissions as an undesired output can be divided into two categories: single-factor carbon emission efficiency and full-factor carbon emission efficiency. Measuring the carbon emission efficiency of single factors is relatively straightforward. It involves assessing the carbon dioxide emissions generated by economic production activities, with the relevant variables of these activities reflected by economic indicators such as GDP (gross domestic product) [14] and transportation turnover [15]. However, these economic indicators are output indicators of economic activities and carbon emission is a negative output, so this does not fully align with the economic significance of efficiency. In contrast, the measurement of total factor carbon emission efficiency is relatively complex. Production is a process that involves multiple input factors [16]. Thus, total factor measurement involves integrating the functions of economic factors such as labor, capital, and energy. It is evident that total factor measurement is more comprehensive and practical than single factor measurement, and is more widely used. Total factor carbon emission efficiency can be used with nonparametric DEA(Data envelopment analysis) methods [17–22] or parametric SFA (Stochastic Frontier

Approach) method measures [23–25]. The DEA method is more suitable for dealing with multi-input and multi-output problems and does not need to set specific functions, and is more widely used [26]. The DEA method is an econometric analysis method that considers multiple inputs and outputs in one framework [27]. The representatives of traditional DEA models are CCR and BCC models [28], both of which are radial and angular. The result of this is that all output variables should change in the same direction. Carbon emissions are undesirable outputs. Obviously, the fewer undesirable outputs and the more expected outputs, the higher carbon emission efficiency. Thus, carbon emissions cannot be included in traditional DEA models [29]. As mentioned above, the traditional DEA models are not fit for the measurement of carbon emission efficiency. In order to solve the problem of unexpected output, Tone separated the output of the traditional DEA model, incorporated the expected output and unexpected output into the DEA model in the form of slack variables, and constructed the Unexpected Output SBM Model [30]. On the one hand, the Unexpected Output SBM Model is non-radial and non-angular science; it is constructed based on slack variables [31]. Hence, the unexpected output can be included in the model, and the measurement deviation caused by the radial and angular variations can be avoided. On the other hand, the basic assumption of the Unexpected Output SBM Model is variable returns to scale; that is, each decision-making unit (DMU) can freely change the production scale. Consequently, the calculated efficiency does not contain scope for scale improvement, which reflects the real economic efficiency [32].

The factors Influencing the carbon emission efficiency of railway transportation include transport vehicle structure (proportion of electric locomotives) [33], GDP per capita [34], average railway transport distance, secondary industry ratio [18,35], and technological progress [36], and these have largely been the focus of prior research. Moreover, existing research explores the spatial distribution of railway carbon emission efficiency [5,18,34,35] and analyzes its spatial convergence and spatial spillover effect [37–41]. These studies highlight improvements in the overall efficiency of railway transport, although this differs at the regional scale. Moreover, evidence suggests that the carbon emission efficiency of provincial transportation in China shows a complex network correlation [37,42,43]. Thus, spatial relationships must be considered in order to improve carbon emissions efficiency. As one of the transportation modes, the spatial correlation network of the carbon emission efficiency of railway transportation needs to be further studied.

Based on this literature review, we can identify the existing research gaps. One is the lack of research on the spatial correlation structure in carbon emissions efficiency in Chinese railway transportation. The continuous improvement of the transport network and the regional coordinated development strategy have broken down the regional barriers to the circulation of labor, technology, capital, products, and other elements of production activities [44]. Additionally, these input and output elements present characteristics of the spatial network. Carbon emissions, one of the outputs of production activities, also present a significant spatial correlation structure [45,46]. Here, we recognize that a spatial correlation structure in carbon emissions efficiency in Chinese railway transportation is possible. The other gap is in terms of data usage, where "attribute" data rather than "relationship" data are used by existing studies to perform the spatial analysis of the carbon emission efficiency. The "attribute" data only reflect the characteristics of the region itself, rather than the relationships between regions. Additionally, the consideration of spatial relevance is confined to geographical proximity [47]. These result in an obstacle to identifying the overall structural characteristics of the spatial correlation network of carbon emission efficiency in railway transportation. However, the overall structural characteristic has a decisive influence on the attribute characteristics of regions, making it valuable in the study of spatial correlation [48]. These research gaps restrict the regional coordination and improvement of carbon emission efficiency in railway transportation, thus posing a significant obstacle to China's carbon emission reduction goals.

To address these gaps, we first measured the carbon emission efficiency of the railway transport industry in China. Then, we constructed a gravitational moment matrix of the

spatial correlation network based on the carbon emission efficiency, and we measured and explored three aspects of the structural characteristics of the spatial correlation network, including the overall, individual and spatial clustering aspects. Finally, we used "relationship" data to analyze factors influencing the spatial association network. Additionally, the research process figure is shown in Figure 1.

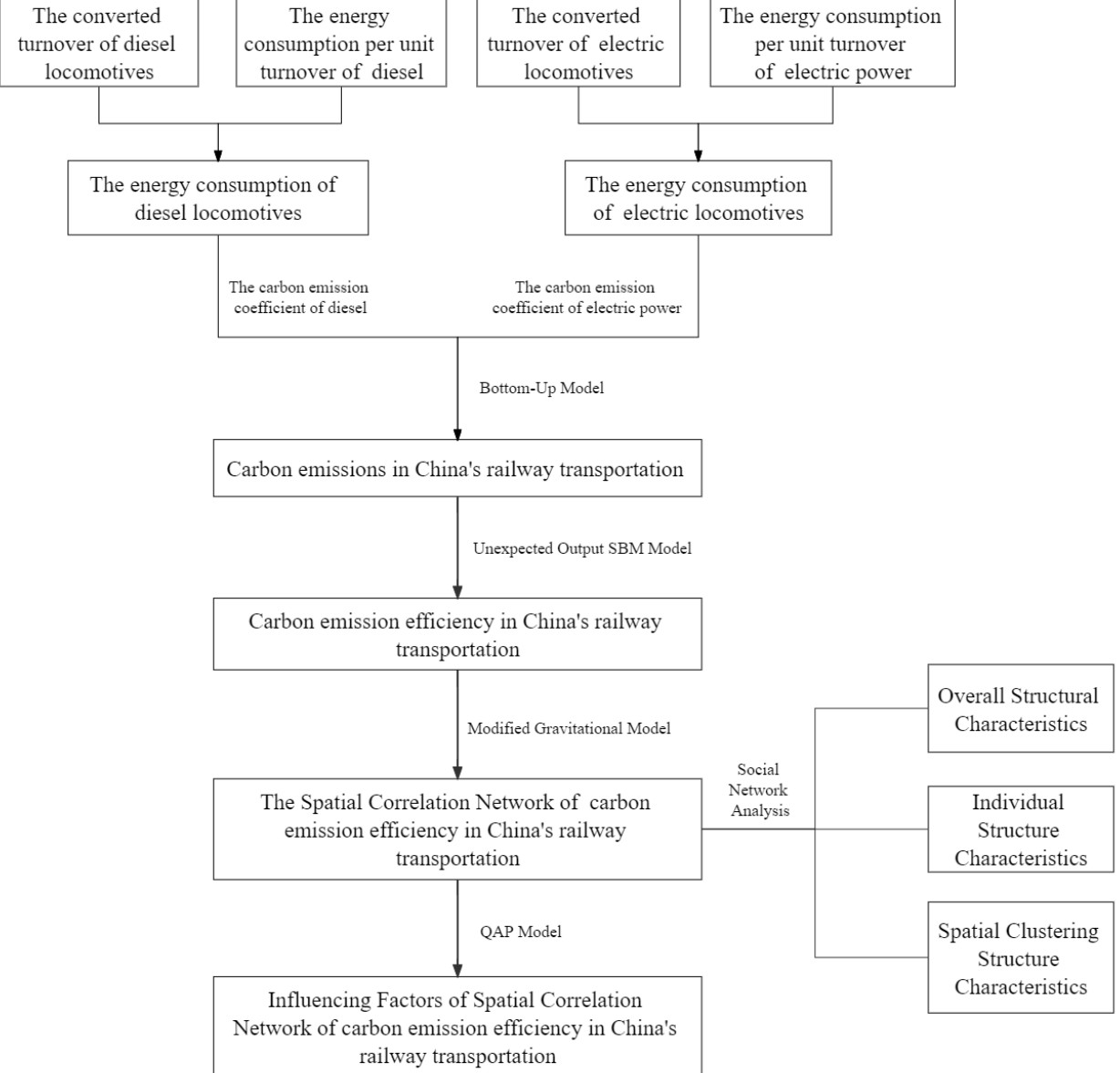

**Figure 1.** The research process.

Compared with existing studies, the marginal contribution of this paper is reflected as follows: (1) Incorporate spatial correlation into the research on railway carbon emission efficiency; (2) Explore the influencing factors of spatial network of carbon emission efficiency in railway transportation through "relationship" data, which better reflect the correlation between regions; (3) Identify the overall structural characteristics of the spatial correlation network of carbon emission efficiency in railway transportation.

The rest of this study is organized as follows: the materials and methods used in this study are described in Section 2. A presentation of the empirical results is provided in Section 3. The discussion is presented in Section 4. The recommendations are presented in Section 5. Finally, the conclusions are presented in Section 6.

## 2. Materials and Methods

### 2.1. Bottom-Up Model

Railway carbon emission efficiency, as an unexpected output, needs to be measured in terms of carbon emissions. Due to the lack of statistical data on railway energy consumption in some provinces, the bottom-up method is selected to calculate railway carbon emissions [49].

(1) Calculation of railway energy consumption

Railway transport mainly includes diesel locomotives and electric locomotives, for which diesel and electric power are the main energy sources. If there are k kinds of railway transport vehicles, the energy consumption of the kth transport vehicle is $E_k$, the converted turnover is $V_k$, and the energy consumption per unit turnover is $F_k$. Energy consumption is obtained by multiplying the converted turnover of different means of transport with the energy consumption per unit turnover, as shown in Equation (1):

$$E_k = V_k F_k, \tag{1}$$

$$V_k = PT_k \times \varphi + FT_k, \tag{2}$$

where $PT_k$ represents passenger turnover, $FT_k$ represents goods turnover, and $\varphi$ represents the conversion factor, the value of which is 1.

(2) Calculation of railway carbon emissions

Railway carbon emissions are calculated based on the consumption of various types of energy and the corresponding carbon emission coefficients. Assuming that $I_k$ is the energy consumption coefficient of the kth transport vehicle, with values of 3.10 kg $CO_2$/kg in the case of diesel and 0.96 kg $CO_2$/kWh in the case of electric power, this can be calculated as follows:

$$C = \sum_{k=1}^{n} C_k = \sum_{k=1}^{n} E_k \times I_k, \tag{3}$$

$$I_k = NCV_k CEF_k COF_k \frac{44}{12}, \tag{4}$$

where $NCV_k$ is the average low calorific value, $CEF_k$ is carbon content per unit calorific value, $COF_k$ is carbon oxidation rate, $C_k$ represents the carbon emission of the $k$th transport vehicle and $n$ represents the total number of types of railway transport vehicles (here, $n = 2$, i.e., diesel locomotive and electric locomotive).

### 2.2. Unexpected Output SBM Model

The unexpected output SBM model is used to calculate railway carbon emission efficiency, which is realized through Max DEA 9 software. Compared with the traditional DEA model, the non-radial and non-angular unexpected output SBM model includes the bad output in the calculation framework and solves the problems of radial increase or decrease in efficiency measurements and the lack of a slack variable in the model. The basic principle of the Unexpected Output SBM Model measuring efficiency is to identify the optimal DMUs from a group of input–output combination DMUs. The production frontier composed of these optimal DMUs is a benchmark, and the efficiency of the remaining DMUs can be calculated by their distance from the corresponding point on the production frontier. Specifically, the production activity of each DMU is represented by a combination of input–output observation data in which carbon emissions are considered as an undesirable output. The points on the production frontier make the most effective use of available resources to obtain the maximum possible expected output. The points outside the production frontier may have insufficient expected output, excessive unexpected output or redundant input, and cannot achieve the optimal efficiency. Assuming that the efficiency of each point on the production frontier is 1, the gap between the efficiency value of each point outside the production boundary and 1 can be quantified according to the distance between each point

outside and the effective point on the production frontier. The relative efficiency can then be obtained.

We selected 31 provinces (excluding Macao, Hong Kong, and Taiwan) as the decision-making units (DMUs). Each DMU has $n$ input variables $X$, $m_1$ expected output variables $Y^a$, and $m_2$ unexpected output variables $Y^b$. The $X$, $Y^a$, and $Y^b$ matrices can be defined as: $X = [x_1, x_2, \ldots, x_N] \in R^{n \times N}$, $Y^a = [y_1^a, y_2^a, \ldots, y_N^a] \in R^{m_1 \times N}$, and $Y^b = [y_1^b, y_2^b, \ldots, y_N^b] \in R^{m_2 \times N}$, $X > 0$, $Y^a > 0$, $Y^b > 0$. The production possibility set is defined as:

$$P = \left\{ \left( x, y^a, y^b \right) \middle| x \geq X\lambda, y^a \leq Y^a\lambda, \sum_1^N \lambda = 1, \lambda \geq 0 \right\} \tag{5}$$

where $\lambda$ represents the weight vector. When the sum of the weight values is 1, i.e., $\lambda \times L = 1$ where $L$ is a unit vector, the production technology belongs to the variable scale reward category; otherwise, it is categorized as a constant scale reward. Therefore, the linear expression of the unexpected output SBM model is given as follows:

$$\rho = min \frac{1 - \frac{1}{n}\sum_{i=1}^n \frac{S_i^-}{x_{i0}}}{1 + \frac{1}{m_1 + m_2}\left( \sum_{r=1}^{m_1} \frac{S_r^a}{y_{r0}^a} + \sum_{r=1}^{m_2} \frac{S_r^b}{y_{r0}^b} \right)}, \tag{6}$$

$$s.t. \begin{cases} x_{i0} = X\lambda + S_i^- \\ y_{r0}^a = Y^a\lambda - S_r^a \\ y_{r0}^b = Y^b\lambda + S_r^b \\ S_i^- \geq 0, S_r^a \geq 0, S_r^b \geq 0, \lambda \geq 0 \end{cases}, \tag{7}$$

where $S_i^-$, $S_r^a$, and $S_r^b$ represent slack variables for the input, the output, and the unexpected output, respectively, and $\rho$ represents the objective function. When $\rho = 1$, $S_i^- = S_r^a = S_r^b = 0$ and the DMU is valid; when $0 \leq \rho < 1$, the DMU is invalid. The input variables include capital, labor, and energy. We selected railway transportation mileage to represent capital input, the number of railway employees to represent the labor input, and energy consumption to represent the energy input. The output variables included expected and unexpected output. The converted turnover was the expected output, while carbon emission was the unexpected output. These indexes are shown in Table 1.

**Table 1.** The indexes of the Unexpected Output SBM Model.

| Index Types | | Index | Data Explanation |
|---|---|---|---|
| Input | Capital | Railway Transportation Mileage | The physical capital accumulated in railway transportation at a certain point in time |
| | Labor | The Number of Railway Employees | Quantity of labor input during railway transportation |
| | Energy | Energy Consumption | Diesel consumed (kg) and electricity consumed (kwh) in railway transportation |
| Output | Expected Output | Converted Turnover | Represents the economic output of railway transportation, reflecting both passenger and freight transport aspects |
| | Unexpected Output | Carbon Emissions | $CO_2$ emissions generated during railway transportation |

*2.3. Modified Gravitational Model*

The Spatial Correlation Network shows the complex network formed by the spatial flow of elements [50]. The spatially separated units act as nodes in the network and interact with each other due to the flow of elements. The element flow constitutes the connection line between nodes [36]. Accordingly, the spatial correlation network is a network of element

interaction. Specifically, the spatial correlation network of carbon emission efficiency is a network composed of the spatial interaction of elements affecting carbon emissions.

As mentioned above, the associations between nodes are the foundation of the spatial correlation network construction. Scholars have largely used the generalized variance decomposition method, which is based on the vector auto regression model (VAR) [51], and the social network analysis method, based on the Gravitational Model, to study spatial correlation networks. The stability of the associations obtained by the VAR is poor due to the strong dependence on lag time [52]. Additionally, it is difficult to characterize the dynamic evolution of the network structure using the VAR [53]. The Gravitational Model, by contrast, works according to the law of universal gravitation and builds associations between network nodes without selecting lagged rank, requires less data, and is relatively simpler to calculate. Moreover, the Modified Gravitational Model takes geographical distance into account in the process of relationship establishment and better describes the dynamic associations between network nodes [54]. This paper, on the basis of the above analysis, decided upon using the Modified Gravitational Model to construct the spatial correlation network of carbon emission efficiency in railway transportation.

Assuming $i$ and $j$ to be two different provinces, the traditional Gravitational Model is as Equation (8), where $f_{ij}$ represents the attraction of railway carbon emission efficiency from object $i$ to $j$. $M_i$ and $M_j$ represent their quality, which is their own quality coefficient. $d_{ij}$ represents the distance between the objects $i$ and $j$. $k_{ij}$ is the gravitational constant. The traditional Gravitational Model is constructed according to the law of universal gravitation and the principle of distance attenuation, which means that the gravity between two objects is proportional to their mass and inversely proportional to the distance between them [55].

$$f_{ij} = k_{ij}\frac{M_i M_j}{d_{ij}^2}, i \neq j, \tag{8}$$

The mass parameters in the traditional Gravitational Model measure the quality of two objects. Additionally, the carbon emission reduction quality of two regions can be measured well using carbon emission efficiency. Additionally, compared with the geographical distance in the original gravity model, the "distance", taking economic differences into consideration, could evaluate the ability to establish associates more objectively [56]. Additionally, based on the characteristics of the data used in this study, differences exist in the interactions of railway carbon emission efficiency among provincial borders of China. Referring to previous studies [57], we introduce the proportion coefficient of total railway transport revenue into the Gravitational Model as a modified empirical constant. Then, assuming $i$ and $j$ to be two different provinces, the gravitational formula of railway carbon emission efficiency between provinces can be calculated as follows:

$$F_{ij} = \alpha_{ij}\frac{Q_i Q_j}{D_{ij}^2}, \alpha_{ij} = \frac{T_i}{T_i + T_j}, D_{ij}^2 = \left(\frac{d_{ij}}{g_i - g_j}\right)^2, i \neq j, \tag{9}$$

where $F_{ij}$ represents the attraction of railway carbon emission efficiency between provinces $i$ and $j$. $Q_i$, $Q_j$, $T_i$, $T_j$, $g_i$, and $g_j$ represent the carbon emission efficiency, total revenue of railway transportation and per capita GDP of the two provinces, respectively. $\alpha_{ij}$ represents the contribution rate of province $i$ to the carbon emission efficiency between the provinces. $d_{ij}$ represents the spherical distance between the provincial capitals. $D_{ij}$ represents the distance between provinces in the gravitational model, which is represented by the ratio of the geographical distance between provincial capitals to the difference in per capita GDP between the two provinces.

We used the Modified Gravitational Model to transform "attribute data" such as carbon emission efficiency, geographical distance, and economic development level into "relational data", which are represented by the correlation intensity matrix of railway carbon emission efficiency for 31 provinces. This correlation matrix clearly observes the spatial

correlation network structure. In order to facilitate the analysis of individual structural characteristics, we transformed the correlation matrix into the binary matrix. We assumed the mean value of each row in the matrix to be the critical value. When it is greater than the critical value, the value of $F_{ij}$ is recorded as 1, which indicates that there is correlation of railway carbon emission efficiency between the two provinces. For the reverse it is recorded as 0, indicating that such a correlation does not exist.

### 2.4. Social Network Analysis

Based on the previous section, we selected the social network analysis method based on the gravity model to explore the structural characteristics of the spatial network. We constructed a spatial correlation network matrix using the modified gravity model through UCINET 6.645 software and drew the topological map of the spatial correlation network using Net Draw 2.161 software. Then, we used the social network analysis method to quantitatively study the overall structural characteristics, individual structural characteristics and spatial clustering structural characteristics of the related network of railway carbon emission efficiency across 31 provinces in China.

### 2.5. Overall Structural Characteristics

We selected four commonly used indicators that characterize the overall network structure, namely, network density, network correlation degree, network grade and network efficiency [37]. The detailed equations for each indicator are given by Liu [51].

Network density reflects the interaction strength of each node in the association network. A large number of relationships in the network increases the network density and results in strong correlations among the provinces. The network correlation degree indicates the accessibility of the associated network. The overall network relevance increases with an increase in the number of directly connected nodes. The network grade reflects the differences in the status of carbon emission efficiency of the various provinces. The stronger the asymmetric accessibility of each node and the greater the number of single relationships, the greater the gradient difference in the network structure. Network efficiency is closely related to the stability of the associated networks. When network efficiency is low, a large number of redundant relationships and overflow channels among nodes exist. In such a scenario, the overall network stability is strong.

### 2.6. Individual Structural Characteristics

To study the individual structural characteristics, we used three indicators, namely, degree centrality, proximity centrality, and intermediary centrality. The detailed equations for each indicator are given by Liu [51]. Degree centrality represents the connectivity between a node and other nodes in the association network. A high degree centrality indicates a large number of direct links between two provinces. Betweenness centrality reflects the degree of control that a node has over other nodes. Betweenness centrality is an indicator of node importance based on the number of shortest paths passing through the node. A node with high betweenness centrality plays the role of an "intermediary" and has great control over information and resources. Proximity centrality indicates the independence of a node. High proximity centrality indicates shorter direct distances between a node and other nodes. Such a node would have greater access to information channels and be less easily controlled by other nodes.

### 2.7. Spatial Clustering Structure Characteristics

White et al. proposed using the block model to first divide the nodes in the spatial association network structure into different plates using the clustering method, and then analyze the roles and interactions of each plate [58]. Wasserman et al. put forward an indexing system for the division of spatial association networks [59]. Here, the role division of each plate was mainly based on the number of internal members, as well as the number of internal and external receiving and sending relationships. The detailed division system is

described in Table 2. Here, it assumed that the number of provinces in the entire association network is g and that a certain section contains $g_s$ provinces.

**Table 2.** Partitioning system of spatial correlation networks.

| Intra-Plate Relations Proportion | Proportion of External Receiving and Sending | |
|---|---|---|
| | $\approx 0$ | $>0$ |
| $\geq (g_s - 1)/(g - 1)$ | Two-way overflow | Net benefit |
| $< (g_s - 1)/(g - 1)$ | Net overflow | Broker |

We built a block model according to the CONCOR method to explore the characteristics of the spatial cluster structure of the spatial correlation network. The process of the CONCOR method is to first calculate the correlation coefficient between each row (or column) of the initial matrix and obtain a correlation coefficient matrix, then to continue to calculate the correlation coefficient between each row (or column) of this correlation coefficient matrix to form a new coefficient matrix, and to repeat the above process many times [51].

We established a density matrix for spatial correlation networks of railway carbon emission efficiency to further explore the relationships among the interpolated values of carbon emission efficiency. If the plate density is greater than the overall density, plates show a trend of clustering and concentration, which is recorded as 1, and the other plates are recorded as 0.

### 2.8. QAP Model

Since the variables used in this paper are all relational data, there may be a high degree of multicollinearity between them. Consequently, traditional methods of measurement may cause large deviations in the regression results. The QAP model does not require the independence of variables and the conformity of interference terms in the normal distribution. We could obtain the regression coefficients of the two variable matrices based on permutation and comparison of the relationship data. Therefore, we used the QAP model, which is realized through the UCINET 6.645 software, to analyze the factors influencing the spatial correlation network structure. We set the random replacement times of UCINET 6.645 software to 5000 times. The spatial correlation network matrix is represented by Q. Based on the domestic research achievements, the equation for the model is as follows:

$$Q = f(D, G, S, R), \tag{10}$$

where $D$ represents the spatial adjacency matrix, $G$ represents the matrix of differences in economic development, $I$ represents the matrix of differences in the railway transport structure, $S$ represents the matrix of differences in industrial structure, and $R$ represents the matrix of differences in scientific and technological advancement. Among these, the level of economic development is measured by the per capita disposable income in each province. The railway transport structure is expressed as the proportion of freight turnover and passenger turnover in each province. The industrial structure is determined by the proportional value of the secondary industry output in the total annual output. The index of scientific and technological level is measured by the research and development (R&D) expenses of each province.

### 2.9. Data Source

The research data represent 31 provinces in China for the period 2006–2019. Taiwan, Hong Kong, and Macao were excluded because of incomplete historical data. The data required for the aforementioned model in this chapter were obtained from the sources shown in Table 3. The geographical distance between the provincial capitals was calculated using ArcGIS 10.7.

**Table 3.** Basic data and their sources.

| Data | Data Source |
|---|---|
| Railway passenger turnover Railway freight turnover GDP per capita Per capita disposable income Output value of secondary industry Total output value | China Statistical Yearbook |
| Number of railway employees Total revenue of railway transportation | China Railway Yearbook |
| Railway transportation mileage | China Transportation Statistics Yearbook |
| R&D expenses | Statistical Bulletin of National Science and Technology Investment |
| Average low calorific value | General Rules for Calculation of the Comprehensive Energy Consumption GB/T 2589-2020 |
| Carbon content per unit calorific value Carbon oxidation rate | 2006 IPCC Guidelines for National Greenhouse Gas Inventories |
| Provinces.shp * | National Fundamental Geographic Information System |
| Border.shp * | |
| Provincial capitals.shp * | |

* Shp (ESRI Shapefile) is a vector graphics format.

## 3. Results

### 3.1. Railway Carbon Emission Efficiency

The data of railway carbon emission efficiency of 31 provinces in China from 2006 to 2019 are shown in Table 4. Additionally, we divided the carbon emission efficiency for the 31 regions into five levels, taking 2006, 2012, and 2019 as examples to draw the spatial distribution map of carbon emission efficiency in China's railway transportation (Figures 2–4). The higher the carbon emission efficiency, the darker the corresponding color.

**Table 4.** Railway carbon emission efficiency of 31 provinces in China from 2006 to 2019.

| | 2006 | 2007 | 2008 | 2009 | 2010 | 2011 | 2012 | 2013 | 2014 | 2015 | 2016 | 2017 | 2018 | 2019 |
|---|---|---|---|---|---|---|---|---|---|---|---|---|---|---|
| Beijing | 0.661 | 0.700 | 0.704 | 0.661 | 0.852 | 1.000 | 0.810 | 1.000 | 0.884 | 0.705 | 0.653 | 0.761 | 1.000 | 1.000 |
| Tianjin | 0.876 | 1.000 | 0.814 | 0.712 | 0.747 | 0.721 | 0.663 | 0.653 | 0.652 | 0.604 | 0.591 | 0.627 | 0.740 | 0.702 |
| Hebei | 0.945 | 1.000 | 0.972 | 0.915 | 1.000 | 1.000 | 0.988 | 1.000 | 0.761 | 0.790 | 0.798 | 0.898 | 1.000 | 1.000 |
| Shanxi | 1.000 | 0.968 | 0.897 | 0.808 | 0.980 | 0.888 | 0.939 | 1.000 | 0.651 | 0.783 | 0.787 | 0.885 | 0.965 | 1.000 |
| Inner Mongolia | 0.806 | 0.758 | 1.000 | 0.915 | 0.922 | 1.000 | 0.949 | 0.902 | 0.708 | 0.659 | 0.611 | 0.756 | 0.917 | 1.000 |
| Liaoning | 0.542 | 0.534 | 0.532 | 0.505 | 0.523 | 0.545 | 0.493 | 0.480 | 0.456 | 0.411 | 0.416 | 0.463 | 0.505 | 0.538 |
| Jilin | 0.410 | 0.401 | 0.408 | 0.395 | 0.413 | 0.441 | 0.425 | 0.414 | 0.397 | 0.360 | 0.365 | 0.414 | 0.461 | 0.486 |
| Heilongjiang | 0.483 | 0.468 | 0.468 | 0.439 | 0.457 | 0.474 | 0.460 | 0.429 | 0.402 | 0.369 | 0.432 | 0.425 | 0.465 | 0.479 |
| Shanghai | 0.358 | 0.294 | 0.285 | 0.275 | 0.267 | 0.264 | 0.267 | 0.274 | 0.289 | 0.303 | 1.000 | 0.343 | 0.383 | 0.409 |
| Jiangsu | 0.494 | 0.433 | 0.389 | 0.378 | 0.370 | 0.386 | 0.404 | 0.431 | 0.441 | 0.447 | 0.437 | 0.490 | 0.549 | 0.491 |
| Zhejiang | 0.451 | 0.463 | 0.458 | 0.403 | 0.413 | 0.412 | 0.403 | 0.397 | 0.381 | 0.396 | 0.429 | 0.446 | 0.510 | 1.000 |
| Anhui | 0.641 | 0.655 | 0.596 | 0.567 | 0.556 | 0.535 | 0.508 | 0.487 | 0.485 | 0.463 | 0.467 | 0.495 | 0.532 | 0.566 |
| Fujian | 0.373 | 0.377 | 0.373 | 0.314 | 0.320 | 0.319 | 0.313 | 0.305 | 0.314 | 0.315 | 0.365 | 0.351 | 0.390 | 0.438 |
| Jiangxi | 0.469 | 0.455 | 0.438 | 0.422 | 0.427 | 0.445 | 0.436 | 0.423 | 0.400 | 0.397 | 0.477 | 0.428 | 0.468 | 0.503 |
| Shandong | 0.666 | 0.637 | 0.620 | 0.615 | 0.625 | 0.619 | 0.576 | 0.549 | 0.493 | 0.467 | 0.475 | 0.505 | 0.530 | 0.594 |
| Henan | 0.639 | 0.641 | 0.628 | 0.623 | 0.602 | 0.629 | 0.586 | 0.583 | 0.551 | 0.526 | 0.520 | 0.576 | 0.628 | 0.656 |

**Table 4.** *Cont.*

|  | 2006 | 2007 | 2008 | 2009 | 2010 | 2011 | 2012 | 2013 | 2014 | 2015 | 2016 | 2017 | 2018 | 2019 |
|---|---|---|---|---|---|---|---|---|---|---|---|---|---|---|
| Hubei | 0.532 | 0.522 | 0.509 | 0.443 | 0.451 | 0.461 | 0.442 | 0.442 | 0.435 | 0.432 | 0.427 | 0.464 | 0.509 | 0.541 |
| Hunan | 0.505 | 0.505 | 0.495 | 0.452 | 0.461 | 0.474 | 0.461 | 0.452 | 0.431 | 0.429 | 0.428 | 0.464 | 0.499 | 0.531 |
| Guangdong | 0.372 | 0.364 | 0.365 | 0.330 | 0.335 | 0.343 | 0.342 | 0.378 | 0.340 | 0.359 | 0.364 | 0.397 | 0.439 | 0.468 |
| Guangxi | 0.651 | 0.654 | 0.629 | 0.586 | 0.610 | 0.606 | 0.588 | 0.521 | 0.482 | 0.427 | 0.420 | 0.443 | 0.477 | 0.513 |
| Hainan | 0.379 | 0.357 | 0.290 | 0.284 | 0.245 | 0.232 | 0.243 | 0.262 | 0.272 | 0.277 | 0.281 | 0.312 | 0.356 | 0.372 |
| Chongqing | 0.347 | 0.354 | 0.360 | 0.375 | 0.376 | 0.380 | 0.367 | 0.361 | 0.352 | 0.353 | 0.365 | 0.381 | 0.428 | 0.459 |
| Sichuan | 0.515 | 0.516 | 0.516 | 0.486 | 0.478 | 0.485 | 0.484 | 0.483 | 0.469 | 0.448 | 0.436 | 0.464 | 0.514 | 0.538 |
| Guizhou | 0.608 | 0.603 | 0.583 | 0.639 | 0.626 | 0.592 | 0.588 | 0.558 | 0.535 | 0.486 | 0.474 | 0.502 | 0.522 | 0.575 |
| Yunnan | 0.435 | 0.478 | 0.463 | 0.447 | 0.458 | 0.464 | 0.474 | 0.487 | 0.478 | 0.474 | 0.458 | 0.484 | 0.514 | 0.545 |
| Tibet | 0.284 | 0.375 | 0.428 | 0.496 | 0.552 | 0.704 | 1.000 | 1.000 | 0.907 | 0.761 | 0.891 | 0.810 | 1.000 | 1.000 |
| Shaanxi | 0.511 | 0.523 | 0.564 | 0.574 | 0.553 | 0.556 | 0.575 | 0.597 | 0.528 | 0.548 | 0.560 | 0.593 | 0.636 | 0.685 |
| Gansu | 0.675 | 0.679 | 0.709 | 0.676 | 0.702 | 0.735 | 0.739 | 0.756 | 0.688 | 0.603 | 0.569 | 0.610 | 0.639 | 0.699 |
| Qinghai | 0.371 | 0.370 | 0.440 | 0.444 | 0.474 | 0.503 | 0.528 | 0.475 | 0.514 | 0.441 | 0.452 | 0.478 | 0.517 | 0.542 |
| Ningxia | 0.749 | 0.730 | 0.726 | 0.735 | 0.730 | 0.739 | 0.783 | 0.725 | 0.623 | 0.568 | 0.561 | 0.594 | 0.594 | 0.569 |
| Xinjiang | 0.583 | 0.585 | 0.623 | 0.551 | 0.550 | 0.535 | 0.524 | 0.526 | 0.514 | 0.468 | 0.454 | 0.509 | 0.575 | 0.627 |
| Mean | 0.559 | 0.561 | 0.557 | 0.531 | 0.551 | 0.564 | 0.560 | 0.560 | 0.511 | 0.486 | 0.515 | 0.528 | 0.589 | 0.630 |

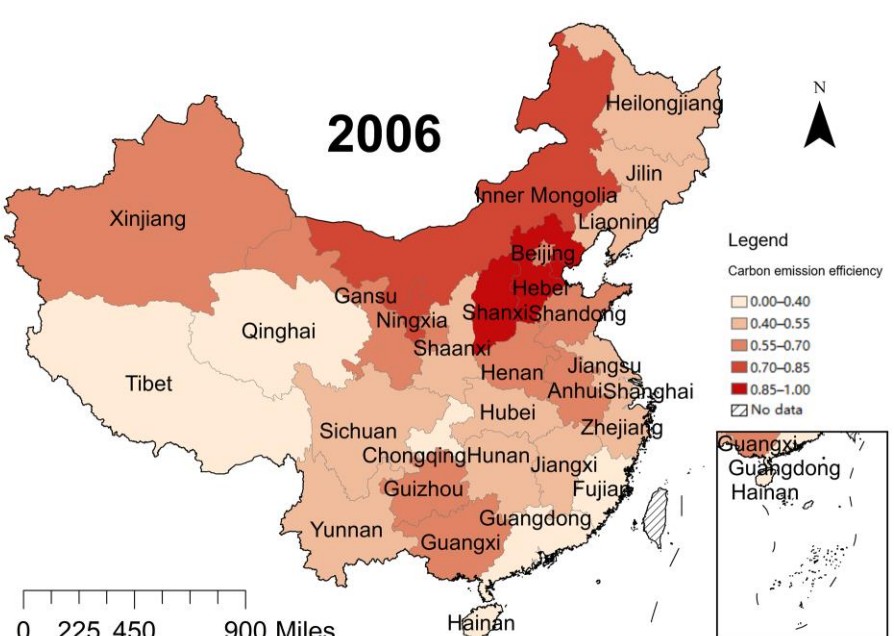

**Figure 2.** The carbon emission efficiency of railway transportation in China (2006).

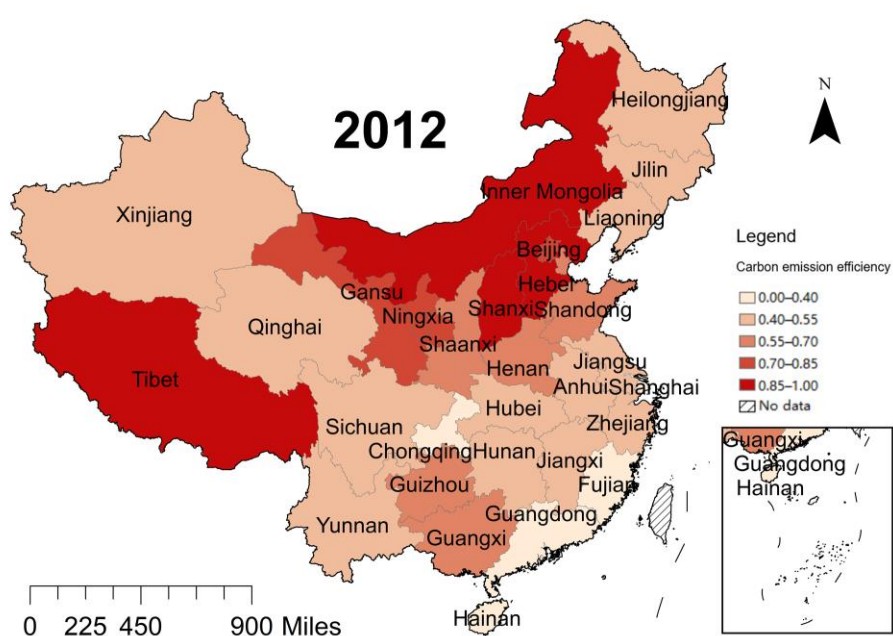

**Figure 3.** The carbon emission efficiency of railway transportation in China (2012).

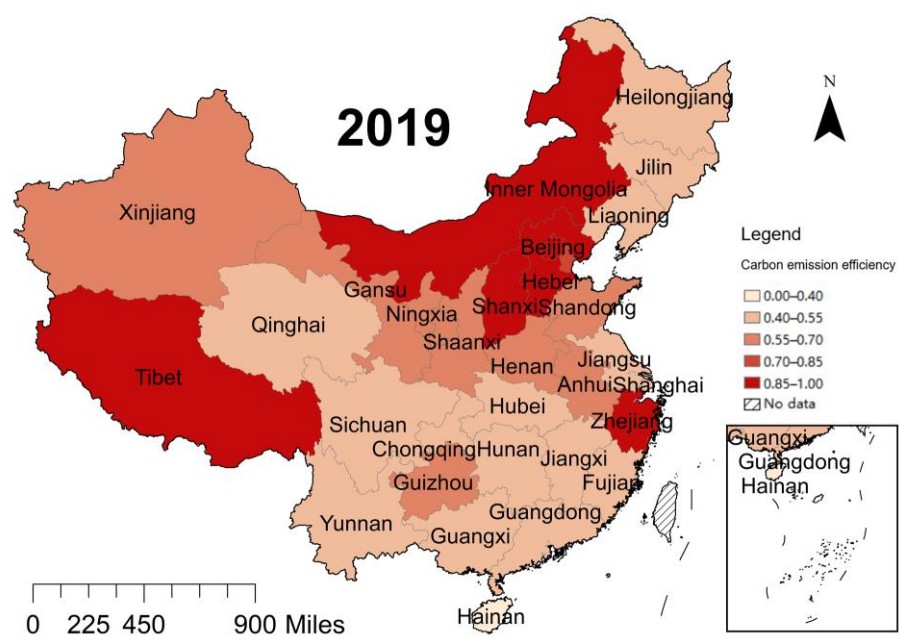

**Figure 4.** The carbon emission efficiency of railway transportation in China (2019).

From the overall perspective, the mean value of railway carbon emission efficiency in 31 provinces in China (Table 4) fluctuated periodically in the range of 0.486–0.630 from 2006 to 2019; it first decreased and then increased. The average values of railway carbon emission efficiency increased from 0.559 to 0.630, indicating that China made some progress in low-carbon development from 2006 to 2019. This means that established inputs and outputs produced fewer carbon emissions, or, the same input produced more output under the constraint of carbon emissions. These average values were less than 1, indicating that from 2006 to 2019 the average railway carbon emission efficiency of 31 provinces in China did not reach the production frontier. The carbon emissions in Chinese railway transportation were still above the theoretical minimum of carbon emissions. Significantly, the average railway carbon emission efficiency began to decline from 2014 and reached its lowest value of 0.486 in 2015. The average value then increased to its peak in 2019.

This is because 2014 and 2015 were the years during which extensive national railway construction was observed. In 2014, 8427 km of new railways were put into production [60]. In 2015, the mileage of new lines put into operation was the highest in the history of China, and the railway operating mileage ranked second in the world [61]. During this period, the resources invested in the railway transportation system were high while the output obtained was low, and then the carbon emission efficiency was low. With technological progress, electric locomotives gradually began to replace diesel locomotives. According to the China Statistical Yearbook, electric locomotives accounted for 57.19% of the total of internal combustion engines and electric locomotives in 2015, and then rose to 62.93% year by year in 2019. Compared with diesel, electric power is cleaner and contributes significantly to carbon emission reduction. The same inputs and outputs have fewer carbon emissions. Therefore, the railway carbon emission efficiency reached a peak in 2019.

From the regional perspective, railway carbon emission efficiency in Beijing, Zhejiang, Hebei, Shanxi, Inner Mongolia, and Tibet reached 1 in 2019 (Table 4). The efficiency in Beijing and Zhejiang was the highest among these, due to their highly evolved economy, advanced railway transportation technology, and strict environmental regulation policies. This ensured greater economic output from the railway transportation system, together with fewer carbon emissions. Hebei, Shanxi, and Inner Mongolia are rich in resources, thus facilitating a dense distribution of railway trunk lines. They transport resources including iron ore, coal, oil, and natural gas all over the country. The turnover of passengers and goods is large, thus achieving a higher output with the constraint of carbon emissions and certain inputs. The development of roads, waterways, and other modes of transportation in Tibet is restricted due to the special terrain and climate. Nevertheless, Tibet is a significant hub for China to establish ties with central and western Asian countries. Therefore, with the established input producing higher passenger and freight turnover, the carbon emissions of railway transport in Tibet are at the theoretical carbon minimum. Additionally, the carbon emission efficiency levels of Shanghai, Fujian, and Hainan provinces were ranked the lowest three in China in 2019. Shanghai and Fujian are both coastal areas; therefore, the majority of freight turnover is in the form of highways and cruise ships. Railway conversion turnover is relatively low. Hainan Province, on the other hand, had a relatively small amount of resources and low cargo turnover and output due to its special geographical location. Moreover, the passenger and freight transport modes used are mainly air and cruise. Consequently, the development of the railway transport industry is low.

From Figures 2–4, carbon emission efficiency of railway transportation in China declined in a gradient from the eastern developed provinces across the central provinces to the western marginal provinces. There was quite a disparity in the carbon emission efficiency of railway transport in the different regions. As time goes by, this disparity has had a reduction tendency. Significantly, the carbon emission efficiency in Tibet rapidly increased. This was because the opening of the Qinghai Tibet Railway in China in 2006 helped to integrate Tibet into China's railway network, greatly reducing the cost of passenger and freight transportation in and out of the country, reducing input, and greatly increasing output. Moreover, the "the Belt and Road" strategy that was first proposed in 2013 gradually became a national strategy and has been comprehensively promoted, with an emphasis on the construction of the land channel in South Asia. In May 2013, the construction of the Bangladesh–China–India–Myanmar economic corridor was completed, thus fully integrating the construction of the Tibet railway network into that of the South Asian land corridor. This increased the passenger and freight volume in Tibet.

### 3.2. Spatial Correlation Network Structure

3.2.1. Overall Structural Characteristics

The network structures of the spatial association networks in 2006, 2012, and 2019 are shown in Figure 5, Figure 6, and Figure 7, respectively.

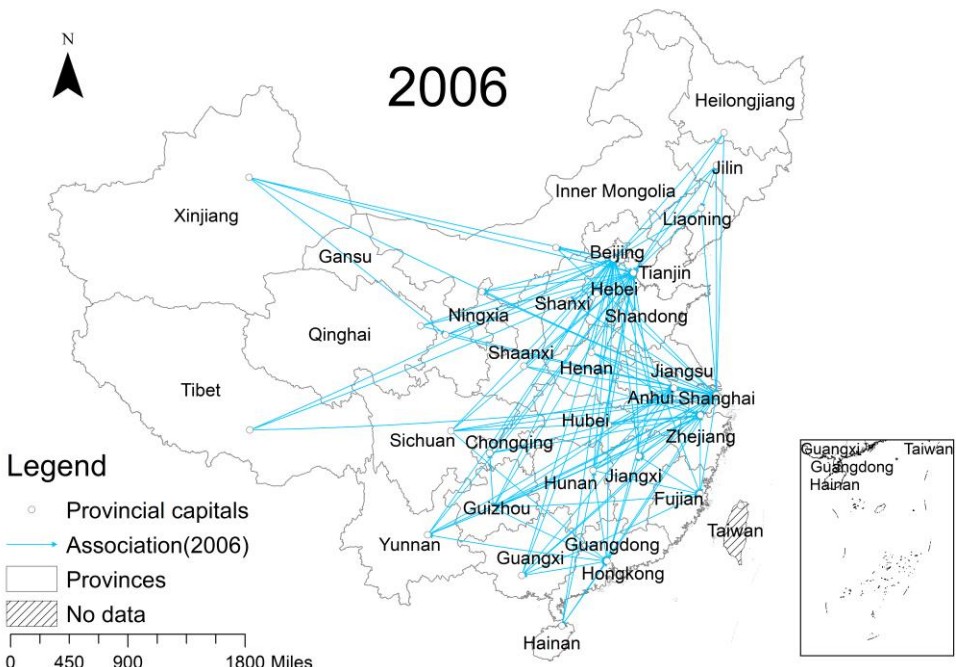

**Figure 5.** Spatial correlation network diagram of railway carbon emission efficiency in 2006.

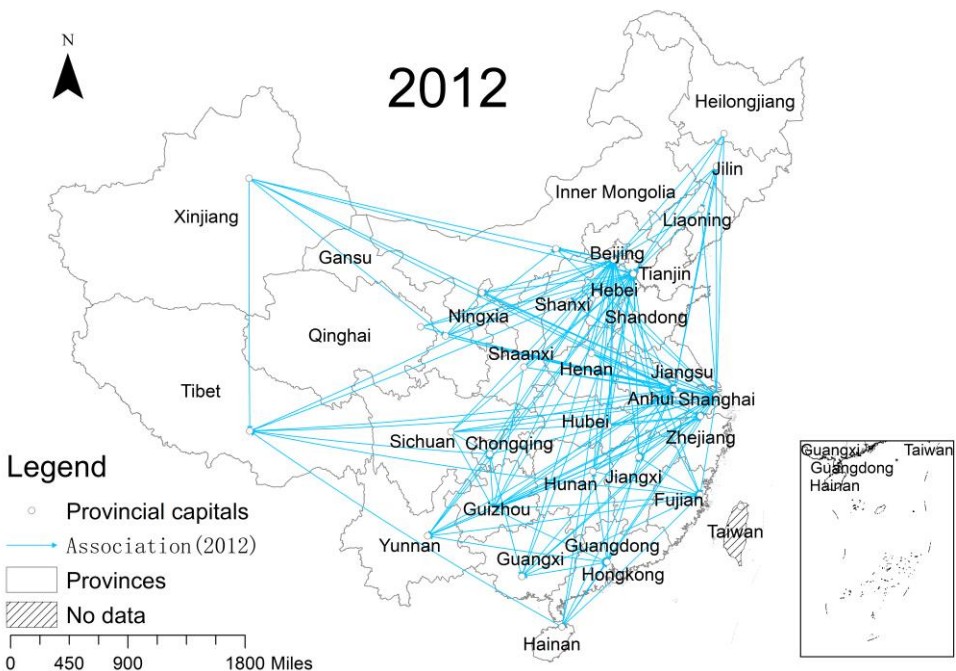

**Figure 6.** Spatial correlation network diagram of railway carbon emission efficiency in 2012.

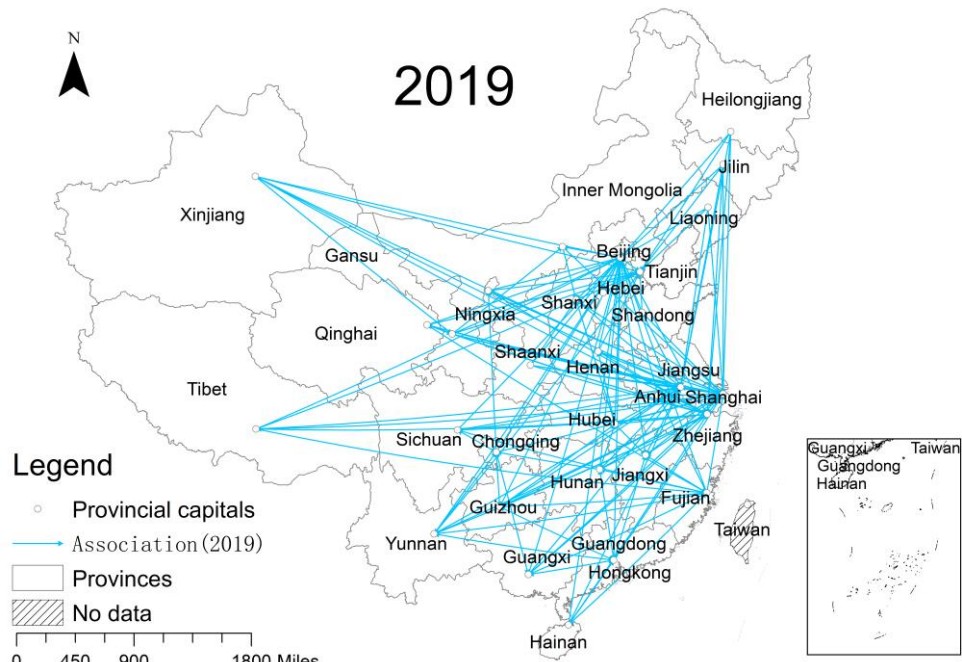

**Figure 7.** Spatial correlation network diagram of railway carbon emission efficiency in 2019.

From these figures, it is evident that the associations between regions that are far apart are no longer restricted by geographical location. From the perspective of network density and network correlation degree, in 2006 there were 140 network relationships and the network density was 0.151. In 2012, the number of network relationships increased to 168, with a network density of 0.181. In 2019, there were 174 network relationships and the network density was 0.187. The number of relationships and the network density of the spatial correlation network show an increasing trend. The overall structure was complex, with an increasing number of connections among the eastern, central, and western provinces. This indicates that the spatial spillover effect has been enhanced. The number of inter-provincial interactions of railway carbon emission efficiency are increasing, resulting in strong complex stability. However, theoretically, the maximum number of network relationships is 930 (31 × 30). There is a large gap between the value of the network density and that of the upper threshold; this indicates that the overall density of the network is low, and the relationships still have great scope to increase. Additionally, the number of spatial correlation network relationships showed a typical declining trend from the eastern provinces across the central provinces to the western provinces in the region (Figures 5–7). There was quite a disparity in the spatial correlation network relationships in different regions. Specifically, the center of the spatial network structure (e.g., Beijing, Shanghai, Tianjin, Jiangsu, and Zhejiang) is highly related to other provinces. In contrast, the edge of the network structure chart (i.e., central and western provinces) has a small number of network relationships.

From the perspective of network grade and network efficiency, the network grade and network efficiency of railway carbon emission efficiency indicated a downward trend throughout the study period. The network grade decreased from 0.538 in 2006 to 0.380 in 2019, indicating the disintegration of the spatial correlation network level gradient. The mutual influence of railway carbon emission efficiency in various provinces had strengthened. Network efficiency decreased from 0.805 at the beginning of the period to 0.751 at the end, indicating that inter-provincial railway connections between nodes in the spatial association network had increased. Interaction relationships between the nodes of the network increased, showing strong network stability. During the research period, the network association of each province was 1, which indicated that no isolated point existed in the spatial association network graph, and that any two nodes could be

reached. The spatial spillover effect was significant, forming a robust spatial association network structure.

Overall, the inter-regional correlation of carbon emission efficiency in railway transportation has been increasing on a yearly basis. Although the spatial association network structure was stable between 2006 and 2019, it was still relatively loose. The correlation relationship gap between the eastern and western regions was significant, and the inter-provincial correlation relationships need to be further increased.

### 3.2.2. Individual Structure Characteristics

Taking 2019 as an example, the individual structural characteristics of spatial correlation network topology based on the railway carbon emission efficiency of 31 provinces are shown in Table 5.

**Table 5.** Spatial correlation network centrality analysis of railway carbon emission efficiency.

| Province | Degree (Centrality) | | | Proximity Centrality | Intermediate Centrality |
|---|---|---|---|---|---|
| | In-Degree | Out-Degree | Centrality | | |
| Beijing | 26 | 6 | 93.333 | 93.75 | 112.339 |
| Tianjin | 25 | 3 | 46.667 | 65.217 | 6.661 |
| Hebei | 2 | 2 | 6.667 | 50.000 | 0.000 |
| Shanxi | 2 | 3 | 23.333 | 56.604 | 30.817 |
| Inner Mongolia | 1 | 3 | 13.333 | 53.571 | 28.833 |
| Liaoning | 1 | 4 | 13.333 | 53.571 | 0.000 |
| Jilin | 1 | 4 | 16.667 | 54.545 | 0.000 |
| Heilongjiang | 0 | 3 | 16.667 | 54.545 | 0.000 |
| Shanghai | 27 | 5 | 90.000 | 88.235 | 192.847 |
| Jiangsu | 10 | 2 | 70.000 | 75.000 | 116.266 |
| Zhejiang | 14 | 3 | 66.667 | 75.000 | 28.128 |
| Anhui | 4 | 5 | 16.667 | 54.545 | 3.000 |
| Fujian | 5 | 4 | 40.000 | 62.500 | 98.897 |
| Jiangxi | 3 | 7 | 23.333 | 56.604 | 128.181 |
| Shandong | 1 | 3 | 13.333 | 53.571 | 0.983 |
| Henan | 0 | 5 | 20.000 | 55.556 | 23.820 |
| Hubei | 0 | 6 | 30.000 | 58.824 | 14.626 |
| Hunan | 1 | 7 | 26.667 | 57.692 | 25.061 |
| Guangdong | 8 | 8 | 26.667 | 57.692 | 78.213 |
| Guangxi | 1 | 7 | 23.333 | 56.604 | 17.616 |
| Hainan | 1 | 5 | 20.000 | 55.556 | 2.094 |
| Chongqing | 1 | 5 | 26.667 | 57.692 | 39.727 |
| Sichuan | 0 | 6 | 20.000 | 55.556 | 1.251 |
| Guizhou | 2 | 8 | 26.667 | 57.692 | 28.880 |
| Yunnan | 1 | 6 | 26.667 | 57.692 | 8.049 |
| Tibet | 0 | 3 | 20.000 | 55.556 | 25.268 |
| Shaanxi | 0 | 4 | 16.667 | 54.545 | 0.000 |
| Gansu | 1 | 3 | 26.667 | 57.692 | 2.879 |
| Qinghai | 0 | 3 | 23.333 | 56.604 | 7.879 |
| Ningxia | 2 | 3 | 16.667 | 54.545 | 1.686 |
| Xinjiang | 0 | 4 | 20.000 | 55.556 | 0.000 |
| Mean | 4.516 | 4.516 | 29.677 | 59.752 | 33.032 |

1.    Degree centrality

The mean value of degree centrality in 31 provinces was 29.677. The highest was Beijing, 93.333, and the lowest was Hebei, 6.667, with a large gap between provinces. In 2019, seven regions, namely, Shanghai, Beijing, Jiangsu, Tianjin, Zhejiang, Fujian, and Hubei, had a higher degree of centrality compared to the mean value of 29.677. This indicates that these regions were at the center of the railway carbon emission efficiency correlation network. They were crucial to the stability of the network structure. This is because these

areas are economically developed and densely populated. With advanced low-carbon railway transport technology and dense transport networks, they have a "siphon effect" and, consequently, a larger number of spatial relations with other provinces. In addition, five of these seven regions, including Beijing, Tianjin, Shanghai, Jiangsu, and Zhejiang, are far more point-in than point-out. These five areas are the national economic development centers. To sustain the rapid economic development, these regions require more technology, capital, and other production resources from other areas. They are in a state of benefit. Eleven provinces (i.e., Jilin, Heilongjiang, Hubei, Hunan, Sichuan, Yunnan, Shaanxi, Tibet, Gansu, Qinghai, and Xinjiang) have significantly higher point-out than point-in. On the one hand, these provinces are located in the northeast or western marginal areas, with lower economic development and more population loss than other regions. On the other hand, most of these provinces are rich in mineral resources. They transport various resources such as coal, oil, natural gas, and nonferrous metals to other regions, resulting in an overflow state.

2.  Proximity centrality

The highest proximity centrality was in Beijing, 93.75, and the lowest was in Hebei, 50, with a mean value of 59.752. In 2019, six regions, namely, Beijing, Shanghai, Jiangsu, Tianjin, Zhejiang, and Fujian, had higher values of proximity than the mean value of 59.752. These provinces have a strong economic foundation, well-developed railway transport networks, high speeds of information transmission, and strong resource acquisition abilities. This indicates that these nodes are close to other nodes in the spatial association network. They enable the easy generation of short path association relationships and are not easily controlled by other provinces. Other regions, such as Heilongjiang, Jilin, Liaoning, Ningxia, Inner Mongolia, Anhui, Shandong and Hebei, had a lower degree of proximity to the center. Due to their relatively marginal geographical location and great distance from other regions, the exchange of railway passenger and freight volume is slow. Consequently, it is difficult for them to contact other regions rapidly.

3.  Intermediate centrality

Shanghai has the highest intermediate centrality, 192.847, and the lowest is 0. This shows that the intermediate centrality of the carbon emission efficiency of inter-provincial railway transportation is characterized by large differences and uneven spatial distribution. In 2019, the top six provinces with intermediate centrality, namely, Shanghai, Jiangxi, Jiangsu, Beijing, Fujian, and Guangdong, dominated the flow of technology, energy, information, and other elements required for the low-carbon development of railway transportation. Further, they regulated the carbon emission efficiency of railway transportation in other regions, as well. They acted as "bridges" and "intermediaries" in inter-provincial relationships. In addition, six provinces, including Hebei, Shaanxi, Liaoning, Jilin, Heilongjiang, and Xinjiang, had an intermediate centrality of 0. They could not act as intermediaries between two nodes in terms of geographical location because of their remote locations in the northeast and the west. Moreover, their low levels of technological and economic development led to a minimal impact on the network relationship, due to which they could not play a "dominant" role.

To sum up, the spatial correlation network of carbon emission efficiency in railway transportation shows an obvious Matthew effect. The above-mentioned indexes of economically developed provinces (e.g., Beijing, Shanghai, Jiangsu, Tianjin, Zhejiang) were higher than those of less developed provinces (e.g., Heilongjiang, Jilin, Liaoning, Xinjiang, Gansu). This shows that the economically developed provinces are at the center of the network, dominating the circulation of elements of low-carbon railway development, while the less developed provinces are at the edge of the network and have a weak ability to obtain development elements.

### 3.2.3. Spatial Clustering Structure Characteristics

Taking 2019 as an example, we built a block model to explore the characteristics of the spatial cluster structure of the spatial correlation network. According to the CONCOR method, the 31 provinces studied were divided into four plates, as shown in Table 6.

**Table 6.** Spatial correlation network division railway carbon emission efficiency in 31 provinces.

| Plate | Province | Receiving Relationships | | Issued Relationships | | Expected Internal Relationship Proportion (%) | Actual Internal Relationship Proportion (%) |
|---|---|---|---|---|---|---|---|
| | | Intraplate | Extraplate | Intraplate | Extraplate | | |
| 1st | Beijing, Tianjin, Shanghai, Jiangsu, Zhejiang | 7 | 101 | 7 | 15 | 13.33 | 31.82 |
| 2nd | Guangdong, Fujian, Chongqing | 0 | 20 | 0 | 23 | 6.67 | 0.00 |
| 3rd | Inner Mongolia, Ningxia, Heilongjiang, Jilin, Liaoning, Hebei, Shaanxi, Shanxi, Gansu, Qinghai, Xinjiang, Tibet, Shandong, Henan, Hubei | 9 | 12 | 9 | 67 | 46.67 | 12.33 |
| 4th | Hunan, Shandong, Anhui, Jiangxi, Sichuan, Guizhou, Guangxi, Hainan | 0 | 25 | 0 | 53 | 23.33 | 0.00 |

From Table 6, it can be observed that in 2019 there were 174 correlation relationships in the network, including 16 within the plate and 158 outside the plate. This explains that the spatial spillover of carbon emission efficiency between plates is significant. The first plate had 7 internal relationships, 15 relationships sent out of the plate, and 101 relationships received outside the plate. Therefore, the first plate is the "main beneficiary plate". In addition, the true internal relationship ratio (31.82%) was higher than the expected ratio (13.33%), indicating that abundant internal connections occurred within the first plate. The second plate sent 23 relationships and received 20 relationships outside the plate domain. The number of relationships sent was approximately the same as the number of those received. Therefore, the second plate is a "broker plate" and acts as a "bridge", with an intermediary role. It is responsible for the transmission of technology, labor, capital, and other factors for reducing carbon emissions or increasing economic output. The third plate, a "net overflow plate", had 9 internal relations, 67 external issued relations, and 12 external receiving relations. The true internal relations ratio (12.33%) was lower than the expected ratio (46.67%). The fourth plate had 25 receiving relationships and 53 spillovers, all from outside the plate. There were no receiving and spillover relationships within the plate, which was also a "net overflow plate". Most of them were resource-exporting provinces.

The overall network density for 2019 was 0.187. We obtained the density matrix and image matrix as described in the Methods section, as shown in Table 7.

**Table 7.** Density matrix and image matrix of spatial correlation plates.

| Plate | Density Matrix | | | | Image Matrix | | | |
|---|---|---|---|---|---|---|---|---|
| | 1st | 2nd | 3rd | 4th | 1st | 2nd | 3rd | 4th |
| 1st | 0.333 | 0.050 | 0.208 | 0.150 | 1 | 0 | 1 | 0 |
| 2nd | 0.600 | 0 | 0.017 | 0.420 | 1 | 0 | 0 | 1 |
| 3rd | 0.938 | 0.033 | 0.083 | 0 | 1 | 0 | 0 | 0 |
| 4th | 0.875 | 0.700 | 0 | 0 | 1 | 0 | 0 | 0 |

From Table 7, it was observed that the first plate had a correlation relationship within the plate domain and received the related elements of carbon emission efficiency which overflowed from each plate. The second plate received the spatial spillover from the fourth plate. The third plate had a spillover effect from the first plate. The spillover effect of the fourth plate acted primarily on the first and second plates. Consequently, the first plate received the spatial carbon emission efficiency overflow from each plate, the second plate was responsible for the transfer of technologies and funds related to low-carbon railway development, and the third and fourth plates mainly spilled the carbon emission efficiency to other plates. The reason for the above phenomenon is that the first plate was located in the Yangtze River Delta and the Capital Economic Circle of China, which is the center of local or national socioeconomic development and at the forefront of low-carbon development of the railway transportation industry. It has a high degree of industrial agglomeration and a high demand for resources. Provinces in the first plate receive resources from other provinces and then play a "leading role" in China's railway low-carbon development. Provincial railway carbon emission efficiency in the second plate is at a moderate level. Therefore, it needs to receive the overflow of advanced low-carbon technologies, talents, and funds from the first plate to improve its own carbon emission efficiency in railway transportation. Meanwhile, the second plate also needs to spill the elements of low-carbon railway development to other plates through the dense railway transport network and inter-sectoral economic links. Thus, it plays a "channel" role. Some provinces in the third and fourth plates are either geographically remote, with poor road network conditions, or regions with large populations and low levels of economic development. Other provinces in the third and fourth plates have vast resource reserves. They transport the necessary labor and resources to other plates, thus playing an "engine" role.

It could be concluded that the eastern region is the main destination for carbon emission efficiency spillovers, and that the western region mainly spills out of the plate.

### 3.3. Influencing Factors of Spatial Correlation Network

### 3.3.1. QAP Correlation

We obtained the correlation coefficient between the spatial correlation network of railway carbon emissions efficiency and various factors influencing carbon emission efficiency in 31 provinces in China (Table 8). Four factors, namely, spatial adjacency, economic development difference, industrial structure difference, and scientific and technological difference, were significant at the 5% significance level. Additionally, railway transport structure difference was significant at the 10% significance level. This indicates that they all have significant relationships with the spatial network. Among these five factors, the coefficients of spatial adjacency, economic development level difference, industrial structure difference and scientific and technological difference were significantly positive. This indicates that they facilitate the spatial correlation of railway carbon emission efficiency values. The coefficient of railway transport structure difference was significantly negative, indicating that similarities in railway transport structure have a catalytic effect on the generation of the spatial correlation of railway carbon emission efficiency in various provinces.

**Table 8.** Correlation analysis of the influencing factors of the spatial correlation network structure.

| Variable | Correlation Coefficient | Significance Level | Mean Value of Correlation Coefficient | Standard Deviation | Min | Max | $p \geq 0$ | $p < 0$ |
|---|---|---|---|---|---|---|---|---|
| Spatial adjacency matrix | 0.155 | 0.000 *** | 0.000 | 0.037 | −0.113 | 0.140 | 0.000 | 1.000 |
| Economic development difference matrix | 0.503 | 0.000 *** | 0.001 | 0.074 | −0.186 | 0.343 | 0.000 | 1.000 |

**Table 8.** *Cont.*

| Variable | Correlation Coefficient | Significance Level | Mean Value of Correlation Coefficient | Standard Deviation | Min | Max | $p \geq 0$ | $p < 0$ |
|---|---|---|---|---|---|---|---|---|
| Railway transport structure difference matrix | −0.085 | 0.081 * | −0.002 | 0.078 | −0.146 | 0.341 | 0.920 | 0.081 |
| Industrial structure difference matrix | 0.178 | 0.021 ** | 0.000 | 0.074 | −0.175 | 0.315 | 0.021 | 0.979 |
| Scientific and technological difference matrix | 0.328 | 0.001 *** | 0.000 | 0.082 | −0.234 | 0.370 | 0.001 | 0.999 |

Note: ***, ** and * indicate that the results are significant at the levels of 1%, 5% and 10%, respectively.

### 3.3.2. QAP Regression

The regression results of the QAP model are shown in Table 9.

**Table 9.** Influencing factors in the regression analysis of spatial correlation network structure.

| Variable | Nonstandard Coefficient | Standardization Coefficient | Significance Level | High Range | Small Range |
|---|---|---|---|---|---|
| Spatial adjacency matrix | 0.259 | 0.230 | 0.001 *** | 0.001 | 1.000 |
| Economic development difference matrix | 0.000 | 0.487 | 0.001 *** | 0.001 | 1.000 |
| Railway transport structure difference matrix | −0.891 | −0.070 | 0.031 ** | 0.988 | 0.013 |
| Industrial structure difference matrix | −0.108 | −0.018 | 0.607 | 0.678 | 0.323 |
| Scientific and technological difference matrix | 0.000 | 0.131 | 0.006 *** | 0.003 | 0.998 |

Note: *** and ** indicate that the results are significant at the levels of 1% and 5%, respectively.

The regression coefficient of the spatial adjacency matrix was significantly positive at the 1% significance level. Geographically adjacent areas have fewer barriers to be correlated, indicating an easy association relationships between geographically adjacent regions.

The regression coefficient of the economic development difference matrix was significantly positive at the 1% significance level, indicating that the gaps in economic development promote the formation of spatial correlation in inter-provincial railway carbon emission efficiency. This is due to the gap in economic development, where the more developed provinces spill over outputs and receive inputs of production activities. This is more likely to drive the economic development of backward areas, thus generating more spatial linkages.

The regression coefficient of the railway transport structure difference matrix was significantly negative at the 5% significance level, indicating that provinces with similar railway transport structures find it easier to form correlations in railway carbon emission efficiency. Similar transport structures indicate similar levels of railway development in these provinces. In such scenarios, the development basis, development requirement and economic output of railway transportation are similar. Inter-provincial low-carbon technological and economic factors are easily circulated, thus strengthening the spatial relationships of railway carbon emission efficiency.

The regression coefficient of the science and technology difference matrix is significantly positive at the 1% significance level, which indicates that differences in scientific and technological advancement are conducive to the establishment of inter-provincial low-carbon development links. Similar to the impact mechanism of differential economic

development on the spatial correlation network, the carbon emission reduction capacity and economic output capacity of railway transportation are closely related to local scientific and technological development. Cities that are more advanced in terms of science and technology find it easy to drive cities with backward science and technology. The advanced provinces provide advanced technology for the backward provinces, while the backward provinces provide resources, human capital, and other essential productive factors for the advanced provinces. Further, the regional coordinated development strategy encourages the exchange and trade of scientific and technological personnel and services between provinces with higher technology levels and those with lower technology levels [62]. This promotes inter-provincial low-carbon links.

The regression coefficient of the industrial structure difference matrix is negative but not significant ($p > 0.1$), indicating that differences in industrial structure do not significantly affect the formation of spatial correlation networks of railway carbon emission efficiency. The industrial structure is expressed as the proportional output of the secondary industry compared to the total output of the industry in a particular province. At present, China has entered the late stage of industrialization. The annual growth rate of the overall manufacturing industry is stable, indicating that China's industrialization is transforming from high-speed development to high-quality development [47]. Therefore, the industrial structure of each province is largely homogeneous and is relatively stable. This leads to a weak impact on railway carbon emission efficiency.

## 4. Discussion

The results obtained in this paper indicate that the carbon emission efficiency in railway transport in China's 31 provinces is gradually improving, and the correlations between provinces are gradually strengthening, but both of these factors show plenty of scope to increase further.

From the perspective of railway carbon emissions efficiency, we first measured the railway carbon emissions efficiency in China's 31 provinces from 2006 to 2019. The average values of railway carbon emissions efficiency in 31 provinces increased from 0.559 in 2006 to 0.630 in 2019, with a cyclical fluctuating trend that first declined and then increased. However, these average values were less than 1. These indicate that though China had made some progress in railway low-carbon development, the carbon emissions in Chinese railway transportation were still above the theoretical minimum of carbon emissions.

From the perspective of the spatial correlation network of railway carbon emissions efficiency, the network relationships increased from 140 in 2006 to 174 in 2019 and the network density increased from 0.151 in 2006 to 0.187 in 2019. The network grade decreased from 0.538 in 2006 to 0.380 in 2019 and the network efficiency decreased from 0.805 at the beginning of the period to 0.751 at the end. These show that inter-provincial railway connections between nodes in the spatial association network increased. However, theoretically, the maximum number of network relationships is 930 (31 × 30) and the upper threshold of network density is 1. Neither the network relationships nor the network density had reached the theoretical maximum value. Thus, the relationships between provinces show plenty of scope to increase further, and the stability of spatial correlation networks in railway carbon emission efficiency needs to be further enhanced.

Meanwhile, we found significant regional heterogeneity in the railway carbon emission efficiency and its spatial correlation structure in China.

From the perspective of the railway carbon emissions efficiency, the spatial distribution map of carbon emission efficiency in China's railway transportation showed significant regional heterogeneity in the eastern and western provinces, indicating that it is difficult for low-carbon railways to develop synchronously in China. The eastern, developed provinces mostly had higher carbon emission efficiency in railway transportation than the central and western provinces. Most of the central and western provinces are economically underdeveloped, without advanced technology to support productive activities. This results in a relatively low output of productive activities and low carbon emission efficiency.

From the perspective of the spatial correlation network of railway carbon emission efficiency, provinces with a high degree of degree centrality, proximity centrality and betweenness centrality belonged to the eastern regions. They dominated the flow of elements required for the low-carbon development of railway transportation, located at the center of the spatial correlation network. Most of the central and western provinces had a low degree of degree centrality, proximity centrality, and betweenness centrality. They were at the edge of the spatial correlation network and had weak control over the flow of elements. Based on the number of internal and external receiving and sending relationships, we divided the nodes in the spatial correlation network into four plates. The four major plates were closely connected and performed their respective duties. The eastern provinces mostly belonged to the first and second plates, receiving the spatial carbon emission efficiency overflow from other plates and overflowing out of the plate at the same time. Most of the central and western provinces belonged to the third and fourth plates, delivering resources needed for railway transportation development to other plates.

Then, we found that five factors, namely, spatial adjacency, economic development difference, railway transport structure difference, industrial structure difference, and scientific and technological difference, had significant correlations with the spatial correlation network of China's railway carbon emission efficiency. Spatial adjacency, economic development difference, and scientific and technological difference can significantly promote the formation of a spatial correlation network for carbon emissions efficiency in the railway transport industry. Similarities in the railway transport structure can promote the formation of a spatial correlation network. Thus, it can be seen that some differences between regions can increase their connections. This finding provides a theoretic basis for policymaking to improve the carbon emission efficiency of railway transportation.

Generally, although there are regional differences in the structural characteristics of the spatial correlation network of railway carbon emission efficiency, interactions between the eastern and western provinces are increasing. Our study provides some valuable routes and methods for the low-carbon development of railway transport: The cross-regional linkages of the low-carbon development of railways are becoming more frequent. The spatial correlation should attach importance to improving the carbon emissions efficiency of railway transport. It should also be noted that there is uneven development in various regions. Different measures should be taken based on the different development levels in different regions.

## 5. Recommendations

Flexible policies are needed to balance economic development and railway carbon reduction. Based on the conclusions we obtained, it is clear that the carbon emission efficiency of China's railway transport industry and its spatial correlation network have obvious regional heterogeneity, and that the network relevance is gradually increasing. Therefore, we make the following policy suggestions based on the principle of regional pertinence and regional relevance.

Firstly, we should improve the inter-regional collaborative mechanism for reducing emissions, as well as inter-regional cooperation and exchange. The "main beneficiary plate" should be encouraged to exchange low-carbon railway technology, resources, and talents with other plates to give full play to its "leading role". The "broker plate" should strengthen the reception of factors from other plates and the transmission of factors to other plates, while further expanding its own scale. The "net overflow plate" primarily belongs to the central and western regions; this makes it necessary to identify the obstacles to both low-carbon development in these areas and communication with other regions, as well as to formulate different policies for the various obstacles. This will create more opportunities for the inter-regional exchange of low-carbon technology, talents, and resources, and remove the obstacles to the establishment of links. An increase in the inter-regional frequency of contact will promote common development.

Secondly, different policies should be formulated for regions with differing levels of development. China's eastern and western railways have different levels of carbon emission efficiency; thus, the government should follow the principle of "common but different". To improve the carbon emission efficiency of railways, provinces such as Beijing, Tianjin, Shanghai, Jiangsu, and Zhejiang, which are in leading positions of the space network, should be prioritized so as to strengthen and allow them to drive the development of other regions. Provinces in the periphery of the space network should actively take measures to develop their economies, reduce carbon emissions, and improve their railway carbon emissions efficiency.

Finally, the differences between regions should be controlled within a certain range. The decrease in railway transport structure difference and spatial distance would increase the connections between provinces. Railway transport structure difference and spatial distance can be reduced by means of inter-provincial railway low-carbon technology exchange, industrial optimization and transfer, and the optimization of railway lines. Further, the increase of differences in economic development, scientific and technological development would increase the spatial correlation of railway carbon emission efficiency. However, excessive differences are likely to lead to the unbalanced development of regional railway transport scales, transport structure, transport speed, etc., increasing the difficulty of establishing the correlation between the two regions. This requires controlling the inter-provincial differences in economic, scientific and technological development within a controllable range, achieving balanced development and obtaining the best results of increasing inter-provincial relations.

To sum up, in view of the characteristics of both China's railway transportation carbon emission efficiency and its spatial correlation network structure, flexible policies should be adopted to reduce railway carbon emissions rather than "one size fits all" policies. On the one hand, we should seek effective ways to increase the correlation between provinces, encourage inter-provincial exchanges, and enhance the stability of the carbon emission efficiency spatial network structure in railway transportation. On the other hand, different policies should be taken in different provinces according to their own characteristics. Additionally, the gaps among provinces should be controlled within a certain range to achieve balanced development.

## 6. Conclusions

In this study, we measured the railway carbon emission efficiency in 31 provinces of China between 2006 and 2019. We established a spatial correlation network for railway carbon emission efficiency at the provincial scale based on the modified gravity model, explored the structural characteristics of the network using the social network analysis method, and identified the factors influencing the network using the QAP model. Then, we drew the following conclusions:

(1) From 2006 to 2019, the railway carbon emission efficiency in these provinces showed a cyclical fluctuating trend that first declined and then increased. The efficiency as a whole still needs to be further improved. Additionally, the provincial railway carbon emission efficiency presents significant regional heterogeneity.

(2) In terms of the spatial correlation network for carbon emission efficiency in railway transportation, the number of inter-provincial relationships is increasing and the correlation of the network nodes is strengthening. However, there are large individual differences between regions. The developed areas in the east, as the center of the network, are vigorously attracting and controlling technology and resources. Additionally, the remote areas in the northeast and the west are at the edge of the spatial correlation network and have weak control over the flow of elements.

(3) Eastern areas are the main members of the "main beneficiary plate" and play a leading role in low-carbon railway development. The east coast, central region and the southwest economic center form a "broker plate" and act as a "channel". Other regions belong to the "net overflow plate", which plays the role of an "engine".

(4) Spatial adjacency, economic development difference and scientific and technological difference can significantly promote the formation of a spatial correlation network for carbon emission efficiency in the railway transport industry. Similarities in the railway transport structure can promote the formation of a spatial correlation network. Although differences in the industrial structure among provinces and regions are also related to the formation of the spatial correlation network, their driving effect is not significant.

Although we have expanded the research perspective and methods on carbon emission efficiency in railway transport, some limitations of this study should be pointed out. Firstly, we have confirmed the influence direction and force of factors influencing the spatial correlation network of railway carbon emissions efficiency, but their specific mechanisms need to be further studied. Secondly, centrality indexes were used to identify the individual structure characteristics of the spatial correlation network. However, the geographical location may affect the accuracy of the results obtained from these centrality indexes. Therefore, the index measurement method needs to be improved to reduce the errors due to geographical location. The above limitations have not been well solved in this study, and need to be paid attention to in further research.

**Author Contributions:** Conceptualization, N.Z., Y.Z. and H.C.; Data curation, N.Z.; Formal analysis, N.Z. and Y.Z.; Methodology, N.Z., Y.Z. and H.C.; Resources, N.Z. and Y.Z.; Software, N.Z. and Y.Z.; Validation, N.Z., Y.Z. and H.C.; Visualization, N.Z.; Writing—original draft, N.Z.; Writing—review and editing, N.Z. and H.C. All authors have read and agreed to the published version of the manuscript.

**Funding:** This research received no external funding.

**Institutional Review Board Statement:** Not applicable.

**Informed Consent Statement:** Informed consent was obtained from all subjects involved in the study.

**Data Availability Statement:** The case analysis data used to support the findings of this study are available from the corresponding author upon request.

**Conflicts of Interest:** The authors declare no conflict of interest.

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
