# Peer review of "Spatial Correlation Network Structure of Carbon Emission Efficiency of Railway Transportation in China and Its Influencing Factors"

_sustainability, doi:10.3390/su15129393_

Round 1
Reviewer 1 Report
1. Explain more about the Unexpected Output SBM Model and the Modified Gravitational Model.
2. Include the research process figure
3. Explain how you prepare and calculate data using ArcGIS 10.7.
4. Optional, include the top 3 states with top and lowest carbon emissions and look at their policies. Perhaps policies are not the problem.
1. Explain more about the Unexpected Output SBM Model and the Modified Gravitational Model.
2. Include the research process figure
3. Explain how you prepare and calculate data using ArcGIS 10.7.
4. Optional, include the top 3 states with top and lowest carbon emissions and look at their policies. Perhaps policies are not the problem.
Reviewer 2 Report
Dear authors,
The conceptual framework, purpose, novelty, academic contribution, literature review, modelling techniques, analysis of the results and discussion, and policy implications of the study are all well written and the paper is worthy for publication.
For readability purposes, only the following points need to be considered for improvement.
Line 21: My recommendation is to replace the word “government” with the phrase “policy makers”.
Lines 201-223: The sub-section “Modified Gravitational Model” needs elaboration. It is not very clear to the reader by which way you integrated the model to the study. For instance, I cannot figure out if the modified gravity model has been estimated, in which format (loglinear, data in levels) and by which estimation method. The abovementioned information could contribute to the easiest interpretation of the results by the reader’s side.
Reviewer 3 Report
Spatial Correlation Network Structure of carbon emission efficiency of railway transportation in China and its influencing factors
Congratulations on your efforts in the preparation of this study. It has an interesting topic, but it needs improvements before the publication decision.
My recommendations are:
1. The literature review should be enhanced with more recent research papers on the field of interest.
2. The results presented in the 'Discussion' section are not enough to explain the main idea of this work; it is recommended to revise the 'Discussion' section.
3. The scientific study will often have a short conclusion, concisely stating the main findings and recommendations for future research, the conclusion section should be revised again.
4. Does the author have any new data on China's railway transportation carbon emissions after 2021?
If yes, it would be nice to add them to the manuscript (see Figure 3 and Table 3).
5. In light of the findings of this study, what recommendations have the authors made? Please add them.
Moderate editing of English language and style required.
Round 2
Reviewer 3 Report
The authors have addressed most of the comments from the previous round, and the paper can be accepted in its current form.
A certificate of editing and proofreading was attached by the authors.